# Cardiovascular Response to Intravenous Glucose Injection during Hemodialysis with Assessment of Entropy Alterations

**DOI:** 10.3390/nu14245362

**Published:** 2022-12-16

**Authors:** Longin Niemczyk, Katarzyna Buszko, Daniel Schneditz, Anna Wojtecka, Katarzyna Romejko, Marek Saracyn, Stanisław Niemczyk

**Affiliations:** 1Department of Nephrology, Dialysis and Internal Diseases, Medical University of Warsaw, Ul. Banacha 1a, 02-097 Warsaw, Poland; 2Department of Biostatistics and Biomedical Systems Theory, Ludwik Rydygier Collegium Medicum, Nicolaus Copernicus University in Toruń, Ul. Jagiellońska 15, 87-067 Bydgoszcz, Poland; 3Otto Loewi Research Center, Division of Physiology, Medical University of Graz, Neue Stiftingtalstrasse 6/V, 8010 Graz, Austria; 4Department of Internal Diseases, Nephrology and Dialysis, Military Institute of Medicine, Ul. Szaserów 128, 04-141 Warsaw, Poland; 5Department of Endocrinology and Isotope Therapy, Military Institute of Medicine, Warsaw, Ul. Szaserów 128, 04-141 Warsaw, Poland

**Keywords:** entropy, end stage renal disease, diabetes mellitus, hemodialysis

## Abstract

Background: The quality of autonomic blood pressure (BP) control can be assessed by the entropy of serial BP data. The aim of this study was to evaluate the effect of hemodialysis (HD) and glucose infusion (GI) on amplitude aware permutation entropy (AAPE) of hemodynamic variables during HD in chronic kidney disease patients with and without type-2 diabetes mellitus (DM). Methods: Twenty-one patients without DM (NDO) and ten with DM were studied. Thirty minutes after the start of HD, a 40% glucose solution was administered. Hemodynamic data were extracted from continuous recordings using the Portapres^®^ system. Results: AAPE decreased during HD in all patients and all hemodynamic signals with the exception of AAPE of mean and diastolic BP in DM patients. GI led to an increase in AAPE for cardiac output in all patients, while AAPE for heart rate and ejection time increased only in DM studies, and AAPE for systolic, diastolic, and mean arterial pressure, as well as total peripheral resistance, increased only in NDO patients. Conclusions: The reduction in entropy during HD indicates impaired autonomic control in response to external perturbations. This state is partially reversed by the infusion of glucose with differences in central and peripheral responsiveness in DM and NDO patients.

## 1. Introduction

The term entropy was introduced to thermodynamics by Rudolf Clausius in 1864 to describe the spontaneity and direction of physical and chemical processes. It is derived from the Greek word “entropía” which refers to “transformation” and “change”. Entropy is a measure of disorder, irregularity, complexity, unpredictability, and information. In this sense, it is of interest in biology, where systems are characterized by variability in spite of overall stability. Constant values indicate a state of low entropy because the state is predictable in future seconds, minutes, or hours [1]. Such a system, however, is ill-equipped to maintain stability when subjected to perturbations.

Variability is especially visible in the cardiovascular system where the interval between successive heartbeats quantified as heart rate variability (HRV) or the amplitude of systolic blood pressures quantified as blood pressure variability (BPV) varies from beat to beat over minutes, hours, and even throughout the day. The variability of these signals is controlled by the autonomic nervous system and importantly affected by disease and components of lifestyle such as exercise, obesity, and smoking. Low variability is a risk factor for cardiovascular disease [2].

Chronic kidney disease (CKD) and diabetic nephropathy are frequent complications of diabetes mellitus [3]. Diabetic neuropathy of the parasympathetic nervous system is manifested by disturbances in HRV and BPV and can lead to syncope, cardiac arrhythmias, and sudden cardiac death, but also to severe hypoglycemia [3]. Disorders of the autonomic nervous system in CKD patients can lead not only to abnormal HRV, including ventricular arrhythmias but also to blood pressure dysregulation and syncope [3,4,5]. Increased sympathetic nervous system activity is highly prevalent in CKD patients treated with hemodialysis, especially in patients with extracellular volume overload [6].

Higher cardiovascular mortality in CKD in comparison with the general population is often explained by autonomic nervous system disorders [7]. Additionally, patients treated with hemodialysis may experience intradialytic syncope, which is also manifested by HRV and BPV abnormalities and which may be due to baroreflex dysfunction, chronic heart failure, impaired vasopressin response, or hypovolemia [8]. Some investigators observed a decrease in sympathetic nervous system activity preceding the drop in blood pressure during hemodialysis, but the results are inconclusive [9,10]. The causes of excessive sympathetic nervous system activation during hemodialysis are not well understood and may include volume expansion and inadequate sodium removal during dialysis [11]. On the other hand, optimizing uremia normalizes HRV [12].

The concept of entropy to quantify the variability of heart rate was introduced by Pincus et al. [13]. Heart rate entropy has been studied in cardiological, orthostatic, exercise, and vasovagal syncope studies. Heart rate entropy has been used to predict atrial fibrillation in spite of an apparently normal sinus rhythm [14,15]. A reduced heart rate entropy has been identified in patients with chronic heart failure [16].

These studies inspired us to analyze the entropy of hemodynamic signals recorded in hemodialysis patients with and without type-2 diabetes mellitus during hemodialysis and glucose-insulin system perturbation by intradialytic administration of glucose.

## 2. Materials and Methods

### 2.1. Study Group

This is the continuation of a study performed in a group of CKD patients with (DM) and without type-2 diabetes mellitus (NDO) treated with hemodialysis in the Dialysis Department of the Military Institute of Medicine in Warsaw, Poland, reported previously [17,18]. More information about material and methods is found in these publications and in companion papers of this study [19,20].

Patients with wasting disease, chronic inflammation, uncompensated anemia, hormone therapy, diabetogenic drug prescription, or inability to provide written informed consent were excluded from the study. All participants in the study were dialyzed using peripheral arteriovenous access without access recirculation and with access flow rates above 600 mL/min, with extracorporeal blood flow during dialysis above 250 mL/min and an adequate dose of dialysis quantified as Kt/V > 1.2.

### 2.2. Study Protocol

The test was conducted in the morning hours. Patients were asked not to eat and drink for more than 3 h prior to starting hemodialysis and during the test and to assume a supine body position for the duration of the test. In DM patients’ insulin was withheld for 8 h before starting the study.

Patients were studied during regular hemodialysis treatment. Thirty minutes after the start of dialysis, a 40% glucose solution was administered at a rate of 0.5 g/kg dry weight (a constant rate of 1 mL/s), and patients were observed for 60 min after the start of the glucose infusion (Figure 1).

### 2.3. Data Preparation and Analysis

Heart rate and arterial blood pressure were noninvasively and continuously recorded from an inflatable cuff mounted on the index or middle finger of the contra-lateral access arm using the Portapres^®^ system (Finapres Medical Systems, Enschede, The Netherlands) with a sampling frequency of 100 Hz from which the following eight signals were extracted: heart rate (HR), systolic blood pressure (fiSYS), diastolic blood pressure (fiDIA), mean arterial pressure (fiMAP), total peripheral resistance (fiTPR), cardiac output (CO), stroke volume (SV), and ejection time (EJT). Data covering a period from 30 min before until 60 min after glucose administration were extracted from the record. Ectopic data were manually excluded leaving more than 95% of the data for further analysis.

For each of the signals, amplitude aware permutation entropy (AAPE) was computed for the phases at the beginning of hemodialysis (start), before and after glucose injection, and at the end of the observation phase (Figure 1). Amplitude aware permutation entropy, which is a variant of permutation entropy (PE) [21], was computed as described previously [22].

Permutation entropy is a measure of complexity of times series based on the order of its values. Simplicity, robustness, and fast computation are the main advantages of PE. It is based on the relationships between values of analyzed time series.

Here we briefly describe the basics of PE calculation according to the description and discussion included in [21,23].

Let us assume that we analyze a time series with length *N*:(1)$${\left\{x_{i}\right\}}_{i=1}^{N}=\left\{x_{1},x_{2},\ldots ,x_{N}\right\}.$$

The embedding of the signal x_i_ in a *d*—dimensional space with delay time *l* gives us a set of reconstruction vectors:(2)$$X_{t}^{d,l}=\left\{x_{t},x_{t+1},\ldots ,x_{t+\left(d-2\right)l,}x_{t+\left(d-1\right)l}\right\},$$
where *t* = 0,1,2…, *N − (d − 1)*. Each $X_{t}^{d,l}$ is arranged in increasing order, as follows:(3)$$\left\{x_{i+(j_{1}-1)l}\leq x_{i+(j_{2}-1)l}\leq \ldots ,\leq x_{i+\left(j_{d-1}-1\right)l}\leq x_{i+\left(j_{d}-1\right)l}\right\}$$
where *j = 1, …, d* is the index of the element in the reconstruction vector. This is a symbolization process. It provides an ordinal pattern $\varphi_{i}=\left\{r_{0},r_{1},\ldots r_{d-1}\right\}$ that describes each vector $X_{t}^{d,l}$. The symbolization process yields in *d*! potential ordinal patterns *π*. The relative frequency of occurrence permutation pattern π_k_ is:(4)$$p\left(\pi_{k}\right)=\frac{\sum_{i=1}^{N-d+1}\delta (\pi_{k^{\prime }}\varphi_{i})}{N-d+1}$$
where *δ*$\left(\pi_{k},\varphi_{i}\right)$ is the Kronecker delta function. The (*δ*$\left(\pi_{k},\varphi_{i}\right)$) is equal to 1 when the ordinal pattern *π_k_* corresponds to the permutation pattern $\varphi_{i}$ and 0 in the other cases.

Permutation entropy (PE) is calculated as:(5)$$PE\left(X,\;d,l\right)=-\sum_{i=1}^{i=d!}p\left(\pi_{i}\right)\cdot \ln\left(p\left(\pi_{i}\right)\right).$$

The maximum PE is achieved when PE reaches ln*(d!)*. Tt occurs when all patterns are equally probable and the analyzed time series is completely random; In case of strictly monotonic signals, PE = 0 and it is a minimal value of PE. Thus, the normalized PE is described by the formula:(6)$$PE\left(X,\;d,l\right)=-\frac{1}{\ln d!}\sum_{i=1}^{i=d!}p\left(\pi_{i}\right)\cdot \ln(p\left(\pi_{i}\right)).$$

The magnitude of PE depends on the embedding dimension *d*; thus, it should be high enough (recommended choosing *d* is between 3 and 7). Nonetheless, the condition *d! << N* should be fulfilled. In our investigation, *d* was set on 7.

The concept of PE is simple and based on a friendly and fast algorithm using a minimal set of parameters. Calculations do not require a long time series. However, this algorithm can be applied to big data sets as well. Unfortunately, the estimation of PE has two key disadvantages: (a) the same permutation pattern could have vectors with totally different amplitudes and average of its element’s values, and (b) differences between amplitude values might not lead to different patterns [22]. Azami and Escudero proposed some corrections to PE to solve the problems mentioned above. Firstly, they added to the calculation of the relative frequency *p*(*π*_k_) the contributions depending on the average absolute (AA) and relative amplitude (RA) [22]. Secondly, they strictly discriminated against ascending/descending sequences [22]. They summed all contributions from patterns representing the same state and divided them by the number of potential permutations of those states.

The formulas for (AA) and (RA) for vector $X_{t}^{d,l}$ are:(7)$$AA_{t}=\frac{A}{d}\sum_{i=1}^{d}\left|x_{t+\left(i-1\right)l}\right|,$$

And
(8)$$RR_{t}=\frac{1-A}{d-1}\sum_{i=2}^{d}\left|x_{t+\left(i-1\right)l}-x_{t+\left(i-2\right)l}\right|,$$

The formula for relative frequency calculation in AAPE method is:(9)$$p^{*}\left(\pi_{k}\right)=\frac{\sum_{t=1}^{N-d+1}\delta \left(\pi_{k},\varphi_{t}\right)\cdot \left(A\cdot AA_{t}+\left(1-A\right)\cdot RA_{t}\right)}{\sum_{t=1}^{N-d+1}A\cdot AA_{t}+\left(1-A\right)\cdot RA_{t}},$$
where *Aϵ[0,1]*. Typically, *A* = 0.5 is recommended; however, depending on the study, one can decide how important is the impact of the changes of amplitude values and average of amplitude values.

AAPE is calculated as follows:(10)$$AAPE\left(X,d,l,A\right)=-\sum_{i=1}^{i=d!}p^{*}\left(\pi_{i}\right)\cdot \ln(p^{*}\left(\pi_{i}\right).$$

The normalized AAPE is given by the formula:(11)$$AAPE\left(X,d,l,A\right)=-\frac{1}{\ln d!}\sum_{i=1}^{i=d!}p^{*}\left(\pi_{i}\right)\cdot \ln(p^{*}\left(\pi_{i}\right)).$$

### 2.4. Statistical Analysis

Distribution of data was assessed by Shapiro–Wilk test. Due to absence of normal distribution, analyses were performed with non-parametric tests. Differences between groups were analyzed by Mann–Whitney test with Holm correction, and differences between time points were analyzed with Friedman test with post-hoc analysis. Mixed models with random effects were built to determine the impact of sex, body mass index (BMI), ultrafiltration volume, ultrafiltration rate, and the concentrations of glucose, sodium, potassium, tonicity and bicarbonate on amplitude aware permutation entropy (AAPE) changes during the examination. The significance level for statistical comparisons was set on α = 0.05 and for the models on α = 0.1, respectively. Continuous data are presented as means with standard deviations. All analyses were performed using Matlab R2020a (Matlab and Statistics Toolbox Release 2020, The MathWorks Inc., Natick, MA, USA) and R version 3.50 (The R Foundation, Vienna, Austria).

## 3. Results

Twenty-one patients without diabetes (NDO), aged 54.95 ± 13.08 years, and ten patients with diabetes (DM), aged 69.70 ± 8.27 years, were included in the study. Patients’ blood glucose concentrations at various time points are presented in a preceding study [18].

Amplitude aware permutation entropy of heart rate (AAPE(HR)) decreased during dialysis (*p* = ns) without difference between DM and NDO groups and increased significantly only in the DM group after glucose injection (*p* < 0.05) (Table 1; Figure 2a). In the mixed model, it was also found that AAPE(HR) increased (*p* < 0.05) with age and tended to decrease with elevated ultrafiltration volume (*p* = ns). Parameters like sex; BMI; tonicity; and plasma levels of glucose, sodium, potassium, and bicarbonate did not significantly change AAPE(HR).

There were no differences in amplitude aware permutation entropy of blood pressure between groups, but hemodialysis reduced AAPE(fiSYS) in both groups (*p* < 0.001), and it significantly changed AAPE(fiDIA) and AAPE(fiMAP) in NDO (*p* < 0.01). AAPE(fiSYS), AAPE(fiDIA), and AAPE(fiMAP) increased in NDO (*p* < 0.001, *p* < 0.01, and *p* < 0.05, respectively) and tended to increase in DM after glucose injection. The entire procedure (dialysis and glucose bolus administration) significantly reduced AAPE(fiSYS), AAPE(fiDIA), and AAPE(fiMAP) (*p* = 0.001, *p* = 0.001, and *p* < 0.001, respectively) (Table 1; Figure 2b–d). It was observed that an increase in the ultrafiltration volume was associated with the rise of AAPE(fiSYS) (*p* = ns), while higher potassium concentration reduced AAPE(fiDIA) (*p* < 0.05). Other parameters, such as sex; age; BMI; ultrafiltration rate; tonicity; and the plasma levels of glucose, sodium, and bicarbonate, did not significantly affect AAPE(fiSYS), AAPE(fiDIA), or AAPE(fiMAP).

Amplitude aware permutation entropy of total peripheral resistance (AAPE(TPR)) was high before and decreased during hemodialysis in both groups (*p* < 0.0001). In the mixed models, time was found to have a significant impact on AAPE(TPR). AAPE(TPR) increased after glucose injection in NDO (*p* < 0.05) and also tended to increase in DM. Completing the entire procedure, including dialysis and glucose bolus administration, AAPE(TPR) decreased in both groups (*p* < 0.0001) (Table 1; Figure 2e). There was no effect of other parameters, such as sex; BMI; ultrafiltration volume; ultrafiltration rate; tonicity; and the plasma levels of glucose, sodium, potassium, and bicarbonate, on AAPE(TPR).

There was no difference in amplitude aware permutation entropy of cardiac output (AAPE(CO)) between groups at all time points, but AAPE(CO) decreased during dialysis (*p* = ns for DM and *p* < 0.05 for NDO). The glucose injection increased AAPE(CO) in both groups (*p* < 0.01 for DM and *p* < 0.05 for NDO) (Table 1). In the mixed model, there was no effect of time or other variables, such as sex; BMI; ultrafiltration volume; ultrafiltration rate; tonicity; and the plasma levels of glucose, sodium, potassium, and bicarbonate, on AAPE(CO).

We found no difference in amplitude aware permutation entropy of stroke volume (AAPE(SV)) between groups at all time points, but AAPE(SV) was reduced by dialysis. The transition from the start of the procedure to the point before glucose injection significantly (*p* < 0.0001) reduced AAPE(SV). There was a tendency to increase AAPE(SV) after the administration of glucose in the DM group, with no change in the NDO group (Table 1). Moreover, lower AAPE(SV) was observed in patients with higher BMI (*p* < 0.05). The other parameters, such as sex; ultrafiltration volume; ultrafiltration rate; tonicity; and the plasma levels of glucose, sodium, potassium, and bicarbonate, did not significantly affect AAPE(SV).

The amplitude aware permutation entropy of ejection time (AAPE(EJT)) decreased during hemodialysis in both groups (*p* < 0.05 for both), and after glucose administration, it significantly increased in the DM group only (*p* < 0.05) (Table 1). It was also observed that AAPE(EJT) declined with increasing ultrafiltration rate (*p* = ns) and potassium concentration (*p* < 0.05) but tended to increase with increasing ultrafiltration volume (*p* = ns). Other variables, such as sex; ultrafiltration volume; tonicity; and the plasma levels of glucose, sodium, and bicarbonate, did not significantly change AAPE(EJT).

## 4. Discussion

In this study, it was found that the amplitude permutation entropy decreased during hemodialysis in all patients with all hemodynamic signals with the exception of AAPE of mean and diastolic pressure in diabetic patients. The infusion of glucose led to an increase in AAPE(CO) in all patients, while AAPE(HR) and AAPE (EJT) increased only in diabetic participants, and AAPE(fiSYS), AAPE(fiDIA), AAPE(fiMAP), and AAPE(TPR) increased only in NDO patients.

The variability and complexity of biological processes are a manifestation of the organism’s potential for adaptation in response to the changing conditions of the environment. Mathematical and statistical methods analyzing the variability and complexity of recorded biomedical signals make it possible to assess these processes and predict the body’s response to specific external stimuli [1]. Studies of the body’s response to specific stimuli are very often reduced to assessing the functioning of the autonomic nervous system and its sympathetic or parasympathetic components. This assessment is made using analyses of heart rate and blood pressure variability and evaluation of changes in their complexity expressed by entropy changes. Variability in entropy rates has been observed both in pathological states and in healthy volunteers, for example, in response to orthostatic stress [24].

Studies show that HRV analysis is suitable for differentiating patients’ conditions and assessing disease severity. In the last 30 years, the attention of researchers involved in the analysis of biomedical signals has been directed to the development of methods that would allow us to evaluate the changes in biomedical signals over time and to use specific parameters to differentiate patients’ conditions. In the 1990s, Pincus proposed and defined entropy for biomedical signals as a measure of signal complexity and called it “approximate entropy”. He alerted researchers to the fact that signals that do not differ in their variability can be characterized by a completely different degree of complexity, which is determined, in the case of the electrocardiogram (ECG), by the sequence in which successive RR intervals occur over time rather than by their averaged value and variance [13]. Pincus’ work has inspired many researchers, both physicists and mathematicians, seeking new approaches to entropy determination, as well as medical practitioners, who see in this new and constantly improving tool the possibility of a more thorough and subtle assessment of changes in patient’s condition and its response to the applied stressor. The use of entropy in ECG signal analysis has resulted in numerous publications on the subject [16,25,26]. Listing the most widely used ones, we must mention sample entropy, conditional entropy, permutation entropy, and transfer entropy. In the research presented in this paper, a modification of permutation entropy proposed by Azami and Escudero called amplitude aware permutation entropy (AAPE) was used to evaluate the complexity of signals [22].

Two systems are thought to be responsible for regulating cardiovascular signal variability: the parasympathetic nervous system (PNS), which is more active at rest and reduces heart rate, and the sympathetic nervous system (SNS), which usually has the opposite effect. However, due to the possible simultaneous activity of both systems, their effect is variable (“teeter-totter”) [27].

Patients with uremia, as well as those with diabetes mellitus, have impaired HRV, which can increase cardiovascular mortality [4,5]. Decreased HRV in patients with chronic kidney disease is an independent predictor of mortality, and the initiation of chronic hemodialysis therapy significantly improves HRV indices [28]. In the study presented here, we found that AAPE(HR) decreased during hemodialysis in both DM and NDO patients without differences between groups, which is consistent with the reports of Rubinger and Tong, but contradictory to the results of Giordano [29,30,31]. Similar trends were observed by researchers in orthostatic tilt tests and explained by an increase in sympathetic nervous system activity and a decrease in its parasympathetic fraction response [24]. In addition, like other authors, we found no effect of electrolyte concentration on entropy changes but observed a decrease in AAPE(HR) with higher ultrafiltration volume [32]. Similar conclusions were reached by other researchers who analyzed HRV and claimed that the baroreflex and peripheral vasoconstriction are responsible for HRV regulation and that normalization of volume status improves HRV but only in patients without diabetes. This may be related to the deterioration of autonomic regulation in DM patients due to diabetic neuropathy [33]. In addition, patients with CKD have increased levels of oxidative stress, which may also cause autonomic dysfunction [34].

The administration of glucose increased APEE(HR), but the increase was only significant in the DM group. The effect of intravenous glucose administration during dialysis on entropy changes in signals recorded from the cardiovascular system has not yet been studied. Some authors have observed an increase in HRV after saline infusion in healthy men during sleep, significantly higher in younger compared to older men, which may indicate an increase in parasympathetic activity in response to acute extracellular volume overload [35].

On the other hand, Brown observed a decrease in resting heart rate and an increase in HRV after oral fluid administration, which was explained by increased baroreceptor sensitivity and elevated cardiovascular stimulation to counteract the rise in blood pressure [36]. In contrast, Christiani found a smaller effect of hypertonic sugar-sweetened sports drinks than water on resting HRV, despite accompanying glucose disturbances [37]. Chapman reported that consumption of sugar-sweetened soft drinks reduced spontaneous baroreceptor sensitivity and heart rate variability but not blood pressure variability [38]. Other authors have noted that insulin stimulates the low- to high-frequency ratio [LF/HF] in spectral analysis of HRV in both lean and obese individuals, thus indicating a shift from the activity of the cardiac autonomic nervous system toward the predominance of the sympathetic nervous system [39].

Some hemodialysis patients experience syncope during hemodialysis and ultrafiltration. Among several reasons for syncope during dialysis, intradialytic hypotension and dialysis imbalance syndrome are enumerated. Intradialytic hypotension is associated with rapid volume depletion. Reduced ultrafiltration and proper assessment of dry weight prevent an excessive fall in blood pressure. Intravenous fluid is usually administered in case of intradialytic hypotension. Dialysis imbalance syndrome is caused by a rapid decrease in plasma tonicity, and the intravenous infusion of osmotically active fluids can improve a patient’s condition and reduce the symptoms by decreasing cerebral edema. [40,41]. In our study, we found that hemodialysis reduced AAPE(fiSYS) in all patients, and it significantly changed AAPE(fiDIA) and AAPE(fiMAP) in NDO subjects. Similar observations of syncope can be observed during tilt tests, and although the mechanism of its onset is probably different, the direct cause of syncope is central hypovolemia [42]. Approaching syncope in a tilt test is usually associated with decreasing entropy of systolic and diastolic blood pressure [25,26,43].

In the tilt test, resuming a supine body position after the orthostatic challenge increases venous return, preload, ventricular filling, stroke volume, and cardiac output. An increase in cardiac output by almost 20% following the infusion of hypertonic glucose in non-diabetic dialysis patients has also been observed in non-diabetic patients on dialysis in a previous study [44]. This increase was explained by a blood volume expansion in the range of 5%. A similar effect can be assumed in this study population, where the blood volume expansion was of comparable magnitude in both NDO and DM patient groups [19]. AAPE(fiSYS), AAPE(fiDIA), and AAPE(fiMAP) increased in NDO and rose in DM after glucose injection, but in DM, this result was not statistically significant.

AAPE(TPR) was high before and decreased during dialysis in both groups and increased after glucose injection. The same effect of dialysis and glucose injection was observed for amplitude aware permutation entropy of cardiac output (AAPE(CO)), amplitude aware permutation entropy of stroke volume (AAPE(SV)), and amplitude aware permutation entropy of ejection time (AAPE(EJT)). If entropy is understood as a measure of dynamic stability and adaptability, the reduction in entropy indicates that hemodialysis compromises the cardiovascular control system to cope with external perturbations and the subject is at higher risk of intradialytic hemodynamic instability. AAPE(TPR) significantly rose after glucose injections in NDO (*p* < 0.05), and it also increased in DM but without statistical significance. Glucose injections caused the rise of AAPE(CO) in both groups (*p* < 0.01 for DM and *p* < 0.05 for NDO). The increase in entropy, therefore, indicates an improvement in cardiovascular control. Moreover, we also reported that EJT increased after glucose injections in DM (*p* < 0.05). These observations, together with the finding that parenteral glucose infusions during hemodialysis increased AAPE(HR) in DM and AAPE of arterial blood pressures in NDO and DM group (*p* = ns), indicate that intravenous glucose injections have the potential to prevent intradialytic hypotension. However, on the other hand, intravenous fluids infusions can increase arterial blood pressure and can induce a major volume perturbation. Shimizu showed that small volumes of rapidly injected hypertonic saline increased blood pressure by stimulating vasopressin secretion [45]. These results were confirmed in the report of Ettema, which found that the rise of plasma sodium during hemodialysis was associated with the increase of plasma vasopressin concentrations [46]. Further studies are needed to clearly identify patients at risk of intradialytic syncope for whom intravenous fluids infusions would be beneficial and would prevent syncope.

As shown in our previous study, intravenous glucose injections during hemodialysis did not worsen the glycemic status in diabetic patients [17]. After intravenous glucose injections, serum glucose concentrations increased significantly, but after one hour of ongoing dialysis, the serum glucose level did not differ from that before hemodialysis in DM patients [18]. Additionally, the intravenous glucose injections did not significantly affect the insulin balance during hemodialysis in patients with diabetes [17].

Based on our results, it seems that AAPE is a promising tool for assessing cardiovascular signals during hemodialysis and during specific interventions such as fluid administration. Future studies could provide additional information on whether AAPE of cardiovascular signals is useful to predict hemodynamic instability so that intradialytic morbid events and the risk of syncope can be avoided by specific countermeasures such as the infusion of concentrated glucose solution. Administration of glucose during hemodialysis, even in people with diabetes, is innovative and safe and can prevent intradialytic hypotension, an important dialysis problem. We applied entropy to analyze the recorded signals in order to assess their complexity.

One of the limitations of our study is the small sample size which did not allow for the inclusion of more confounding factors, such as the type of medications used. Comorbidities may also be another important confounding factor. The observation was limited to 31 patients. Much larger patient numbers in further studies would enable more generalized results.

## 5. Conclusions

In summary, amplitude aware permutation entropy of cardiovascular signals decreases during hemodialysis, indicating reduced adaptability of cardiovascular control mechanisms to respond to this therapeutic system perturbation. The decrease in entropy is reversed by glucose administration, indicating an improvement in cardiovascular control mechanisms. The magnitudes of the changes in entropy depend on the type of cardiovascular variable and on the group of patients studied. A blunted response to hemodialysis and to glucose injections in DM patients is probably due to impaired autonomic system function.

## Figures and Tables

**Figure 1 nutrients-14-05362-f001:**
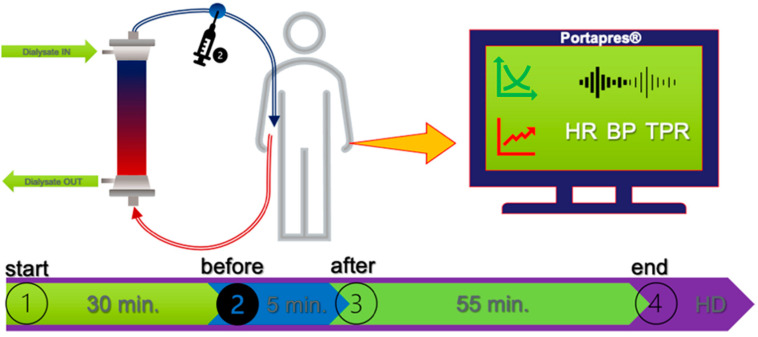
Study protocol.

**Figure 2 nutrients-14-05362-f002:**
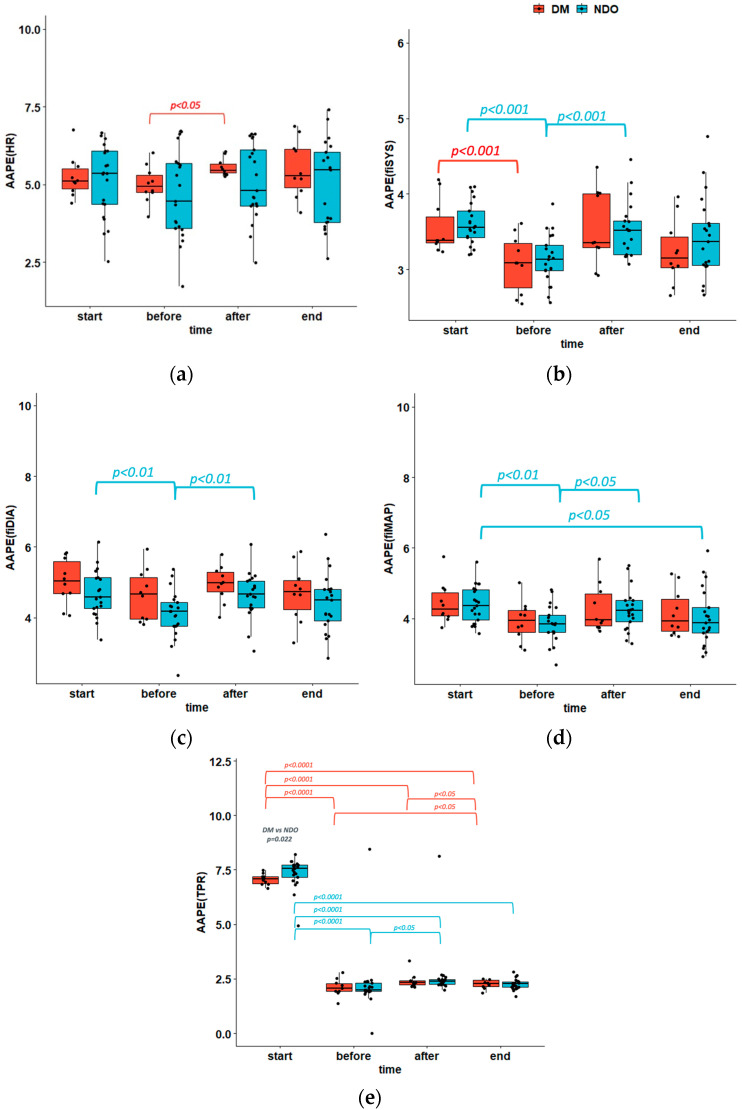
Amplitude aware permutation entropy of (**a**) Heart rate, (**b**) Systolic blood pressure, (**c**) diastolic blood pressure, (**d**) mean arterial pressure, (**e**) Total peripheral resistance.

**Table 1 nutrients-14-05362-t001:** Amplitude aware permutation entropy for different variables compared between groups at different time points.

	Before HD (t = −30)	Before Glucose Injection (t = 0)	After Glucose Injection (t = 5)	End of Test (t = 60)	*p* *
AAPE(HR)	*p* = ns **	*p* = ns **	*p* = ns **	*p* = ns **	
DM	5.25 ± 0.66	5.00 ± 0.59	5.56 ± 0.28	5.51 ± 0.92	*p*_4_ < 0.05
NDO	5.10 ± 1.17	4.66 ± 1.42	5.08 ± 1.22	4.97 ± 1.40	*p* = ns
AAPE(fiSYS)	*p* = ns **	*p* = ns **	*p* = ns **	*p* = ns **	
DM	3.55 ± 0.36	3.07 ± 0.38	3.55 ± 0.50	3.23 ± 0.42	*p*_1_ < 0.001
NDO	3.60 ± 0.28	3.14 ± 0.33	3.53 ± 0.36	3.42 ± 0.54	*p*_1_ = *p*_4_ < 0.001
AAPE(fiDIA)	*p* = ns **	*p* = ns **	*p* = ns **	*p* = ns **	
DM	5.02 ± 0.65	4.64 ± 0.73	4.96 ± 0.52	4.70 ± 0.79	*p* = ns
NDO	4.69 ± 0.67	4.10 ± 0.68	4.63 ± 0.64	4.42 ± 0.83	*p*_1_ = *p*_4_ < 0.01
AAPE(fiMAP)	*p* = ns **	*p* = ns **	*p* = ns **	*p* = ns **	
DM	4.43 ± 0.59	3.92 ± 0.57	4.29 ± 0.68	4.16 ± 0.66	*p* = ns
NDO	4.40 ± 0.52	3.84 ± 0.52	4.24 ± 0.59	4.04 ± 0.79	*p*_1_ < 0.01 *p*_3_ = *p*_4_ < 0.05
AAPE(fiTPR)	*p* < 0.05 **	*p* = ns **	*p* = ns **	*p* = ns **	
DM	7.07 ± 0.25	2.11 ± 0.39	2.40 ± 0.36	2.26 ± 0.20	*p*_1_ = *p*_2_ = *p*_3_ < 0.0001 *p*_5_ = *p*_6_ < 0.05
NDO	7.34 ± 0.69	2.27 ± 1.50	2.64 ± 1.27	2.26 ± 0.25	*p*_1_ = *p*_2_ = *p*_3_ < 0.0001 *p*_4_ < 0.05
AAPE(CO)	*p* = ns **	*p* = ns **	*p* = ns **	*p* = ns **	
DM	2.27 ± 0.17	2.10 ± 0.21	2.32 ± 0.12	2.29 ± 0.14	*p*_4_ < 0.01
NDO	2.28 ± 0.14	2.33 ± 1.40	2.53 ± 1.09	2.29 ± 0.22	*p*_1_ < 0.05 *p*_4_ < 0.05
AAPE(SV)	*p* = ns **	*p* = ns **	*p* = ns **	*p* = ns **	
DM	3.36 ± 0.58	2.96 ± 0.44	3.41 ± 0.50	3.23 ± 0.55	*p*_1_ < 0.0001
NDO	3.47 ± 0.48	3.14 ± 0.52	3.33 ± 0.53	3.47 ± 0.49	*p*_1_ < 0.0001
AAPE(EJT)	*p* = ns **	*p* = ns **	*p* = ns **	*p* = ns **	
DM	7.07 ± 0.25	6.69 ± 0.40	7.13 ± 0.25	7.01 ± 0.44	*p*_1_ = 0.05 *p*_4_ = 0.05
NDO	7.34 ± 0.69	6.69 ± 1.39	6.87 ± 1.28	7.07 ± 0.85	*p*_1_ = 0.05

* *p* for difference in group: *p*1 = (t-30/t0), *p*2 = (t-30/t5), *p*3 = (t-30/t60), *p*4 = (t0/t5), *p*5 = (t0/t60), *p*6 = (t5/t60). ** *p* for DM vs. NDO. The statistical analyses were conducted with Mann–Whitney test with Holm corrections and Friedman test with post-hoc analysis. Abbreviations: DM—patients with diabetes mellitus, NDO—patients without diabetes mellitus, AAPE(HR)—amplitude aware permutation entropy of heart rate, AAPE(fiSYS)—amplitude aware permutation entropy of systolic blood pressure, AAPE(fiDIA)—amplitude aware permutation entropy of diastolic blood pressure, AAPE(fiMAP)—amplitude aware permutation entropy of mean arterial pressure, AAPE(fiTPR)—amplitude aware permutation entropy of total peripheral resistance, AAPE(CO)—amplitude aware permutation entropy of cardiac output, AAPE(SV)—amplitude aware permutation entropy of stroke volume, AAPE(EJT)—amplitude aware permutation entropy of ejection time.

## Data Availability

The data presented in this study are available on request from the corresponding author. The data are not publicly available due to Polish General Data Protection Regulation.

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
