# Peer review of "Cardiovascular Response to Intravenous Glucose Injection during Hemodialysis with Assessment of Entropy Alterations"

_nutrients, 2022, doi:10.3390/nu14245362_

Round 1

Reviewer 1 Report

The article aimed to assess the impact of hemodialysis (HD) and glucose injection upon entropy of heart rate and blood pressure during HD session in end stage renal disease (ESRD) patients with and without diabetes mellitus (DM) type 2.  In 31 HD patients (21 without DM and 10 with DM) the authors found that the Amplitude Aware Permutation Entropy tends to decrease during HD, and glucose injections increase AAPE(HR) in DM patients.

The topic is very interesting but to clarify the result I need more information:

1) It may be useful to describe the population with more clinical data. Some information may be helpful in understanding the outcomes such as: patients' drug therapy (antihypertensive drug and/or cardiology therapy), vascular access (AV fistula and CVC), patients' KT/V.

2) It is interesting to know the blood glucose value of the patients before the glucose infusion and after the glucose infusion (baseline-30'-60').

3) The authors could study the AAPE (HR) in the same patients during HD sessions without glucose infusion to understand the relative role of HD and glucose infusion on the outcome shown

4) It is important to give a clinical message, without this, it is difficult to understand the clinical relevance of the observation. To this aim, it may be useful to describe whether different variations of AAPE(HR) identify different patients with different clinical risk of intradialytic  hypotension or  long term mortality

Author Response

Thank you for your review. I would like to respond to your comments and suggestions (there are in 

Reviewer 2 Report

The article considers the impact of hemodialysis and glucose injection which is evaluated by analyzing the entropy of heart rate and blood pressure during hemodialysis in patients with and without diabetes mellitus. Which are analyzed with an Amplitude Aware Permutation Entropy - This study shows that glucose injection increases HR and blood pressure in DM patients. Generally, the topic is interesting, however, the paper's novelty is vague.

My suggestion:

The study included 31 subjects, so it may not be statistically significant.

Also, a related works section should be added.

Could you please write the paper organization by the end of the introduction section?

Could you please add more details regarding the basic clinical characteristics of the respondents? What are 10 subjects with DM?

Could you please highlight the novelty and contribution in the abstract in introduction?

Author Response

Thank you for your review. Answers to your comments and suggestions are in attached file.

Round 2

Reviewer 1 Report

Thanks to the authors for their replies, I have no further revisions to make